# Convergence between the microcosms of Southeast Asian and North American pitcher plants

Leonora S Bittleston[1,2]*, Charles J Wolock[1,2], Bakhtiar E Yahya[3], Xin Yue Chan[4], Kok Gan Chan[4,5], Naomi E Pierce[1,2], Anne Pringle[6]

[1]Department of Organismic and Evolutionary Biology, Harvard University, Cambridge, United States; [2]Museum of Comparative Zoology, Harvard University, Cambridge, United States; [3]Institute for Tropical Biology and Conservation, Universiti Malaysia Sabah, Kota Kinabalu, Malaysia; [4]Division of Genetics and Molecular Biology, Institute of Biological Sciences, Faculty of Science, University of Malaya, Kuala Lumpur, Malaysia; [5]International Genome Centre, Jiangsu University, Zhenjiang, China; [6]Departments of Botany and Bacteriology, University of Wisconsin-Madison, Wisconsin, United States

**Abstract** The 'pitchers' of carnivorous pitcher plants are exquisite examples of convergent evolution. An open question is whether the living communities housed in pitchers also converge in structure or function. Using samples from more than 330 field-collected pitchers of eight species of Southeast Asian *Nepenthes* and six species of North American *Sarracenia*, we demonstrate that the pitcher microcosms, or miniature ecosystems with complex communities, are strikingly similar. Compared to communities from surrounding habitats, pitcher communities house fewer species. While communities associated with the two genera contain different microbial organisms and arthropods, the species are predominantly from the same phylogenetic clades. Microbiomes from both genera are enriched in degradation pathways and have high abundances of key degradation enzymes. Moreover, in a manipulative field experiment, *Nepenthes* pitchers placed in a North American bog assembled *Sarracenia*-like communities. An understanding of the convergent interactions in pitcher microcosms facilitates identification of selective pressures shaping the communities.

DOI: https://doi.org/10.7554/eLife.36741.001

*For correspondence: leobit@gmail.com

Competing interests: The authors declare that no competing interests exist.

## Introduction

Similar selective pressures in geographically distant habitats can cause unrelated organisms to converge in both morphological and functional traits. Pitchers of carnivorous plants have evolved repeatedly and independently to have the same shapes and insect-trapping functions in Southeast Asia, North America, and Australia (*Albert et al., 1992*). Similar selective pressures can also cause the independent emergence of multispecies interactions with parallel physiological or ecological functions, defined as 'convergent interactions' (*Bittleston et al., 2016b*). The concept of convergent interactions was developed in detail in *Bittleston et al. (2016b)*, and can be used as a tool to better understand forces influencing interspecific relationships. Here, we investigate whether convergent interactions can be identified between different, independently evolved pitcher plant genera and the arthropods and microbes housed within their pitchers. We hypothesize that the microbial communities formed within the fluids of distantly-related pitcher plant species possess similar community structures and functions, and we test this hypothesis by comparing the bacterial and eukaryotic communities living within the plant-held waters (phytotelmata) of pitcher plants from two genera in

**eLife digest** The ecosystems found across the Earth, including in forests, lakes and prairies, consist of communities of plants, animals and microbes. How these organisms interact with each other determines which ones grow and thrive. We still do not understand how communities form: why different species exist where they do, and what enables them to survive in different locations. This knowledge is particularly limited with regard to communities of microbes because they are hard to see and count.

Pitcher plants are an ideal system for studying how communities and ecosystems assemble. The pitcher-shaped leaves of these plants each contain small aquatic communities of microbes and arthropods (including insects and mites) that can be relatively easily studied. Because unrelated groups of plants have evolved pitchers at different times and on different continents, these communities can also be used to explore how evolutionary history and the current environment determine which species thrive in a particular location.

Bittleston et al. sampled the DNA of the communities living within 330 pitchers from various North American and Southeast Asian pitcher plant species. This revealed that very distantly related plants on opposite sides of the planet have pitchers that host similar communities, with the organisms found in one pitcher plant often closely related to the organisms found in others. The genes within the community's DNA also shared many functions, with the majority of shared genes devoted to digesting captured insect prey. Bittleston et al. also relocated pitcher plants from Southeast Asia to grow alongside North American species and found the same microbes and arthropods colonizing both groups, indicating that the different types of pitchers present a similar habitat.

Overall, the results of the experiments performed by Bittleston et al. suggest that certain kinds of interactions between species (such as between the pitcher plants and their microbes) can evolve independently in different parts of the world. Researchers can use these interactions to learn more about how communities and ecosystems form. With a greater understanding of the Earth's ecosystems, it will be easier to protect them and predict how they will fare as global conditions change.

DOI: https://doi.org/10.7554/eLife.36741.002

different plant orders, *Nepenthes* (family Nepenthaceae, order Caryophyllales) native to Southeast Asia and *Sarracenia* (family Sarraceniaceae, order Ericales) native to North America.

Microbial communities are complex, and even apparently simple habitats house orders of magnitude more microbial species than plant or animal species (*Horner-Devine et al., 2004*). Parsing the principles shaping microbial community structure and function remains a challenge. But virtually all plants and animals interact with microbes (*van der Heijden et al., 2008*; *McFall-Ngai et al., 2013*), and host morphology and ecology are likely to shape associated microbial communities. Convergent plant or animal forms, chemistry, or habitats may control the communities within them (*Bittleston et al., 2016b*), analogous to a kind of 'extended phenotype.' Convergently evolved organisms are ideal systems for understanding how different aspects of form or ecology influence microbial communities because they enable distinctions among the effects of evolutionary history, geography, and host morphology or physiology.

A nascent understanding of how microbial communities assemble in association with highly specialized, convergently evolved hosts is emerging. Interactions between animals and their gut microbiomes are frequently mediated by diet. For example, animals with fermenting foreguts as distantly related as the hoatzin bird and cow are colonized by microbial communities with similar structures (*Godoy-Vitorino et al., 2012*), and the gut bacteria of disparate animals whose diets consist of ants also possess a convergent community composition (*Delsuc et al., 2014*). In turn, the herbivorous, arboreal ants of different taxonomic groups are associated with specific bacteria likely to supplement their low-nitrogen diets (*Sanders et al., 2017*; *Hu et al., 2018*). And convergent shifts in bacterial communities are observed in different cichlid fishes as diets change from herbivory to carnivory (*Baldo et al., 2017*). Similar convergent dynamics are observed in sea urchin larvae (*Carrier and Reitzel, 2018*). Animals' microbial associates are likely to impact fitness. The pitchers of

carnivorous pitcher plants are the plant analog of an animal gut, and may function in digestion of animal prey. Convergently evolved genera of pitcher plants provide a tractable model and opportunity to explore the assembly and functional potentials of host-associated microbial communities.

In fact, pitcher microcosms have long served as elegant models for investigating metacommunities and community assembly (*Kneitel and Miller, 2003*; *Buckley et al., 2003*; *Srivastava et al., 2004*; *Armitage, 2017*; *Bittleston, 2018*); each pitcher pool is a discrete but similar habitat with a unique history. In our study, we take advantage of the pitcher system to focus primarily on comparisons of the communities of convergently evolved plants, testing how unrelated hosts with similar morphologies and functions shape their associated communities. Implicit within our comparisons are myriad community assembly mechanisms and processes, including dispersal and environmental filtering caused by characteristics of the host.

Two genera of carnivorous pitcher plants, *Sarracenia* and *Nepenthes*, evolved independently in North America and Southeast Asia, respectively (*Albert et al., 1992*). Pitchers are highly modified leaves, and both genera grow pitchers to attract, trap, and digest insects—primarily ants. Pitcher microcosms also house communities of bacteria, fungi, protozoa, and arthropods (*Beaver, 1983*; *Bradshaw and Creelman, 1984*; *Kitching, 2000*). Pitchers of both genera have characteristic shapes, although forms can vary greatly in color and size among different species. Pitchers of both genera also offer extra-floral nectar to attract prey, possess slippery interiors to trap prey, and secrete digestive enzymes to break down prey tissues (*Juniper et al., 1989*; *Adlassnig et al., 2011*; *Kurup et al., 2013*). *Nepenthes* species produce more different kinds, and a greater abundance, of enzymes compared to *Sarracenia* species, and *Sarracenia* species may rely more on their bacterial communities for prey degradation (*Butler et al., 2008*; *Moran and Clarke, 2010*; *Baiser et al., 2011*). Pitchers actively absorb nitrogen, phosphorus, and other elements from prey; these nutrients are otherwise scarce in the soils where the plants grow (*Chapin and Pastor, 1995*; *Schulze et al., 1997*; *Ellison, 2006*). While the plants are perennial, individual pitchers can last from a few weeks to two years, depending on the species, and are generally most active for the first few weeks to months after opening (*Heard, 1998*; *Osunkoya et al., 2008*). Pitcher interiors appear to be sterile before opening (*Peterson et al., 2008*; *Buch et al., 2013*) (but see [*Chou et al., 2014*]), and once open, a complex community forms within (*Beaver, 1983*; *Kitching, 2000*; *Koopman et al., 2010*; *Krieger and Kourtev, 2012*; *Chan et al., 2016*). Many pitcher-associated organisms are specialists, and are restricted to the pitcher habitat for at least part of their lives (*Fish and Hall, 1978*; *Beaver, 1983*). Various arthropods have co-diversified with their pitcher plant host, suggesting ecological dependence and a shared evolutionary history (*Satler and Carstens, 2016*). In fact, even though the species *S. purpurea* was introduced to Europe over 100 years ago, it houses very few insect inquilines as compared with pitchers in native habitats (*Zander et al., 2016*); the close associations of pitchers and arthropods may be slow to evolve.

To test for convergence between the microcosms of North American *Sarracenia* and Southeast Asian *Nepenthes*, we collected fluids from over 330 pitchers of six species of *Sarracenia* and eight species of *Nepenthes* from native habitats in the United States, Singapore, and Borneo (*Supplementary file 1* Table S1), and used next generation sequencing to characterize the biodiversity housed in each pitcher. To the best of our knowledge, ours are the first comparisons of the entire communities, encompassing bacteria, microbial eukaryotes, and arthropods, associated with convergently evolved organisms; our sampling is also more intensive than any sampling previously published for pitcher plants. We tested for convergence between communities by comparing species richness, community composition, phylogenetic structure, and functional potential. We hypothesized the living communities housed in unrelated pitchers would converge, both structurally and functionally: tests of the hypothesis would result in similar species richness, phylogenetic structure, and functional potential between the *Sarracenia* and *Nepenthes*, as compared to the same parameters measured from surrounding bog water and soil communities. In addition to describing the bacterial and eukaryotic communities of each of the 14 different species of pitcher plants, we explored which features of host species appear to drive patterns of biodiversity. Finally, in a field manipulation, we experimentally tested whether North American insects and microbes would colonize Southeast Asian *Nepenthes* pitchers when *Nepenthes* plants were placed in a North American *Sarracenia* habitat; the experiment tests whether the pitchers of different genera function as similar selective environments when exposed to the same microbial pool. In the aggregate, data provide evidence

for the convergence of pitcher microcosms between independently evolved host genera, and identify aspects of pitcher form and physiology underpinning the similarities.

## Results

### The species richness and evenness of North American and Southeast Asian pitcher plant microcosms converge

To compare the microbial communities within *Sarracenia* and *Nepenthes* pitchers, we analyzed DNA samples from pitchers and their surrounding environments using an amplicon sequencing approach, separately characterizing bacteria and eukaryotes (*Figure 1A*, *Supplementary file 1* Table S2, *Supplementary file 2* Dataset S1). Communities from Southeast Asian and North American pitchers were defined as converging if the communities were more similar to each other than to the communities of the environments immediately surrounding the plants, even despite the vast geographic distance between them. In fact, the *Nepenthes* and *Sarracenia* pitcher communities were distinct from and had fewer Operational Taxonomic Units (OTUs, clustered at 97% sequence similarity; a proxy for species) than communities in surrounding bog water or soil (*Figure 1B and C*, and *Supplementary file 1* Table S3). The pattern held for both bacteria and eukaryotes, and was unaffected by sample volume (no correlation of observed OTUs with sample volume; for bacteria: R = −0.003, p=0.984, and for eukaryotes: R = −0.003, p=0.812). Pitcher samples also had significantly lower Shannon diversities (Mann-Whitney U Test, p<0.001 in all comparisons) than surrounding environments (*Figure 1B*), and this pattern also held when we controlled for extraction volume (by analyzing a subset of 155 samples, each extracted from the same volume; *Supplementary file 1*

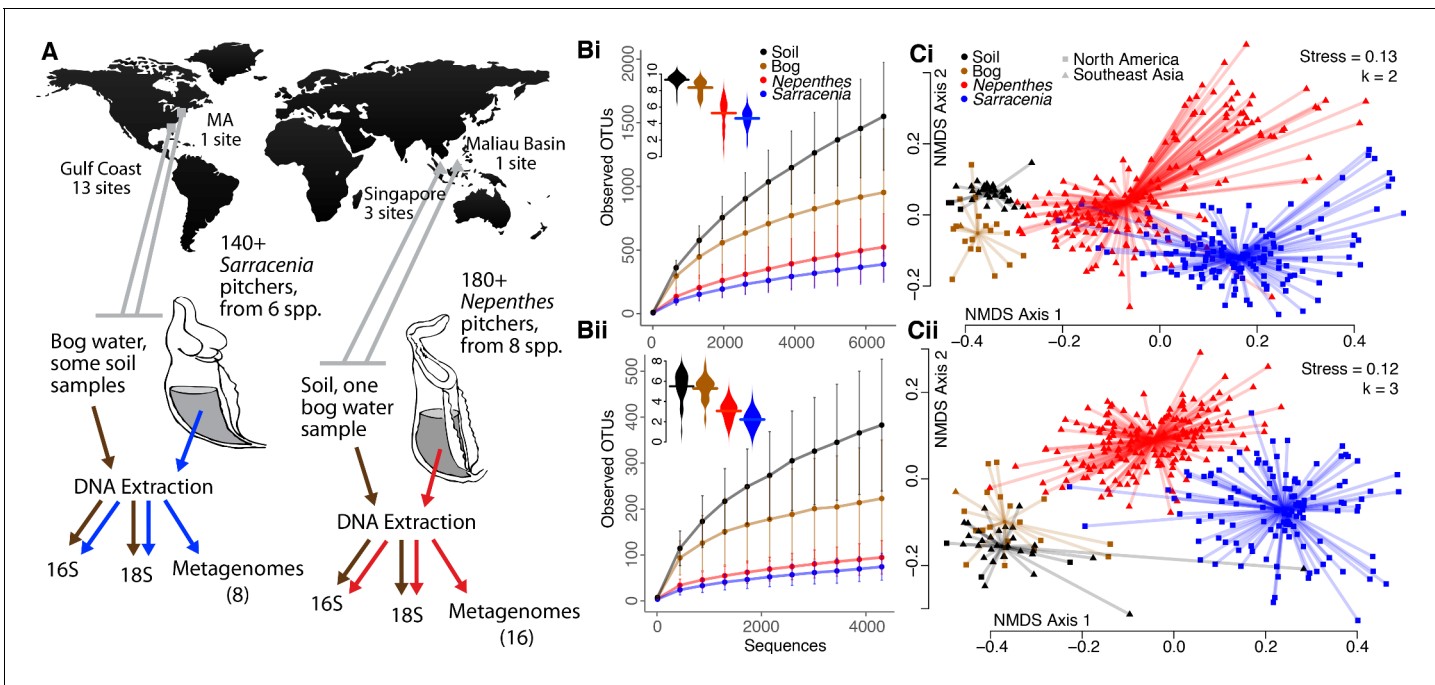

**Figure 1.** Pitcher microcosms are more similar to each other than they are to communities of surrounding habitats. (A) Geography of sampled *Sarracenia* and *Nepenthes* and experimental approach. (B) The species richness (displayed as rarefaction plots) and Shannon diversity (inset beanplots) of both bacterial (i, top) and eukaryotic (ii, bottom) communities was lower in pitchers than in surrounding soil and bog water. Error bars are standard deviations. (C) Community composition using the unweighted UniFrac metric for bacteria (i, top) and eukaryotes (ii, bottom). NMDS stress and dimensions (k) are listed, and the center of each cluster is the category's median value.

DOI: https://doi.org/10.7554/eLife.36741.003

The following figure supplement is available for figure 1:

**Figure supplement 1.** Southeast Asian pitcher communities are more similar to the communities living in water captured in fallen leaves or experimental tubes than those of soil or bog water.

DOI: https://doi.org/10.7554/eLife.36741.004

Table S3). Overall, pitcher communities were characterized by both decreased richness and evenness as compared to communities from their immediate environments.

The composition of pitcher communities was also significantly different from the community composition of surrounding bog water or soil (*Figure 1C*, *Supplementary file 1* Table S4. Bacteria: *envfit*: $R^2 = 0.31$, p<0.001, *adonis*: $R^2 = 0.08$, p<0.001; Eukaryota: *envfit*: $R^2 = 0.38$, p<0.001, *adonis*: $R^2 = 0.08$, p<0.001). To understand differences in community composition across just one region, we separately tested and analyzed Southeast Asian samples from pitchers, bog water, soil, plastic tubes, or cupped, dead leaves filled with water and sitting on the ground. The communities in water from leaves or from plastic tubes were more similar to pitcher fluid communities than bog water or soil communities (*Figure 1—figure supplement 1*).

## Organisms found in *Nepenthes* and *Sarracenia* pitcher microcosms are typically from the same phylogenetic clades

To compare the phylogenetic structures among pitcher communities, we mapped OTUs present in at least 10% of our field (not experimental) *Nepenthes* or *Sarracenia* samples onto bacterial and eukaryotic phylogenetic trees, together with all OTUs found in bog water and soil (*Figure 2* and associated *Figure 2—figure supplement 1*). Organisms repeatedly colonizing *Nepenthes* or *Sarracenia* pitchers in North America and Southeast Asia tended to be from similar clades of bacteria or eukaryotes (*Figure 2*). The pattern was most pronounced in bacteria, and shared families included Microbacteriaceae, Gordoniaceae, Chitinophagaceae, Sphingobacteriaceae, Bradyrhizobiaceae,

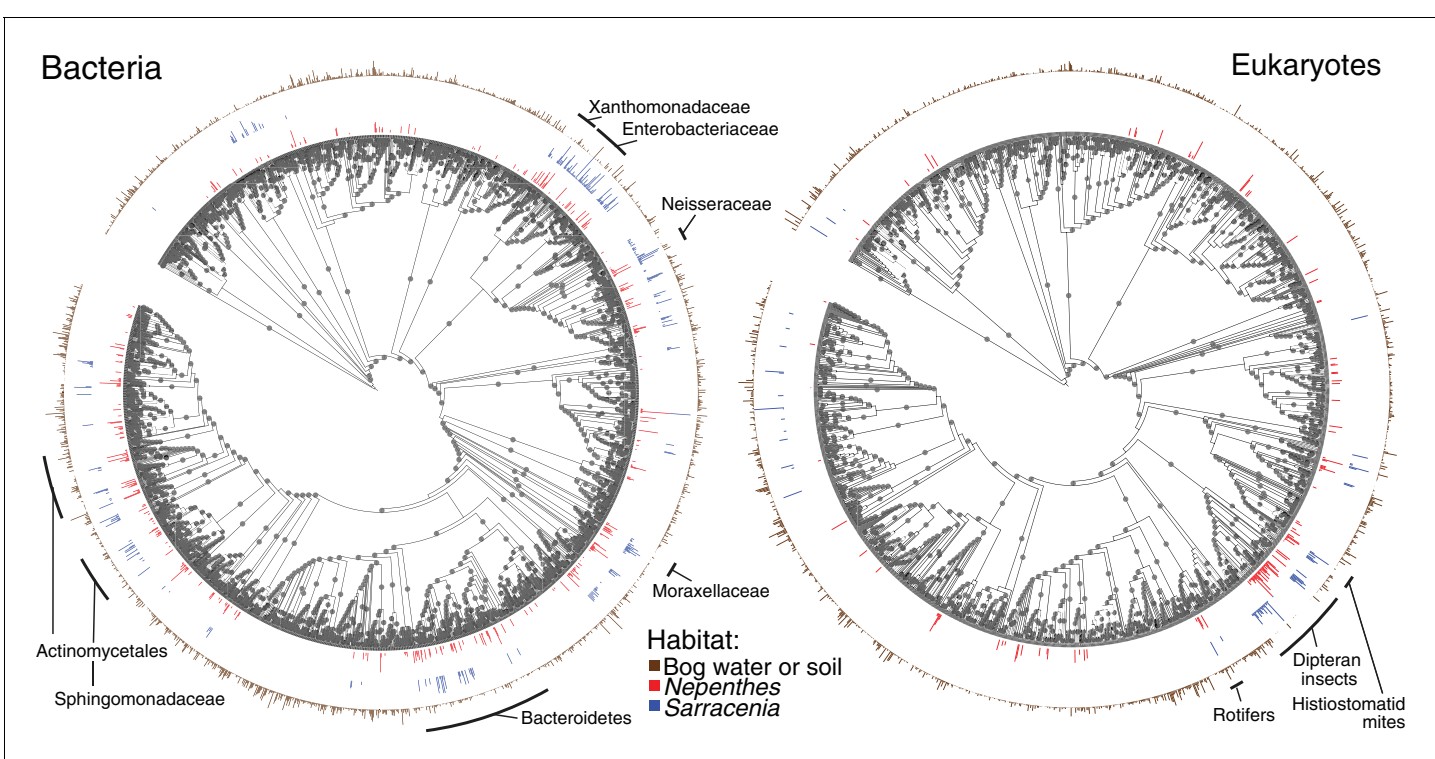

**Figure 2.** *Nepenthes* and *Sarracenia* pitchers are colonized by related organisms. Phylogeny of bacterial and eukaryotic OTUs found in soil and bog samples (brown); and OTUs present in at least 10% of field collected *Nepenthes* (red) or *Sarracenia* (blue) samples. The height of the colored bars surrounding each tree corresponds to the natural log of the relative abundance of reads from each OTU, normalized across the samples in each category. Labels designate monophyletic clades where high proportions of OTUs are shared between *Nepenthes* and *Sarracenia* samples. Gray dots mark branches leading to nodes with bootstrap support of 0.7 or higher. The bacterial tree is rooted in Archaea and the eukaryotic tree is rooted in Streptophyta.

DOI: https://doi.org/10.7554/eLife.36741.005

The following figure supplement is available for figure 2:

**Figure supplement 1.** The same phylogenies as in *Figure 2*, with branch lengths included and branches colored by taxonomic assignment.

DOI: https://doi.org/10.7554/eLife.36741.006

Rhizobiaceae, Sphingomonadaceae, Burkholderiaceae, Comamonadaceae, Oxalobacteriaceae, Neisseriaceae, Enterobacteriaceae, Moraxellaceae, and Xanthomonadaceae. Across eukaryotes shared clades included dipteran insects, mites, and rotifers.

## Within each genus, pitcher species, acidity, form, and volume correlate with community composition

To investigate drivers of community composition among pitchers of each genus, we had recorded species identity and measured the pH and total volume of pitcher fluid associated with each sample. As we extracted DNA from each sample, we recorded DNA concentrations; a proxy for the living biomass within a pitcher (*Marstorp et al., 2000*). In both the *Sarracenia* and *Nepenthes* systems, pitcher communities differed significantly among host species (*Figure 3*, *Supplementary file 1* Table S4). For bacteria, the effect of host species was similar in *Sarracenia* (*envfit* $R^2$ = 0.40, p<0.001; *adonis* $R^2$ = 0.14, p<0.001) and *Nepenthes* (*envfit* $R^2$ = 0.38, p<0.001; *adonis* $R^2$ = 0.18, p<0.001) species; however, for eukaryotes, pitcher host species explained more of the observed variation in *Sarracenia* species (*envfit* $R^2$ = 0.42, p<0.001; *adonis* $R^2$ = 0.20, p<0.001) as compared to *Nepenthes* species (*envfit* $R^2$ = 0.22, p<0.001; *adonis* $R^2$ = 0.15, p<0.001).

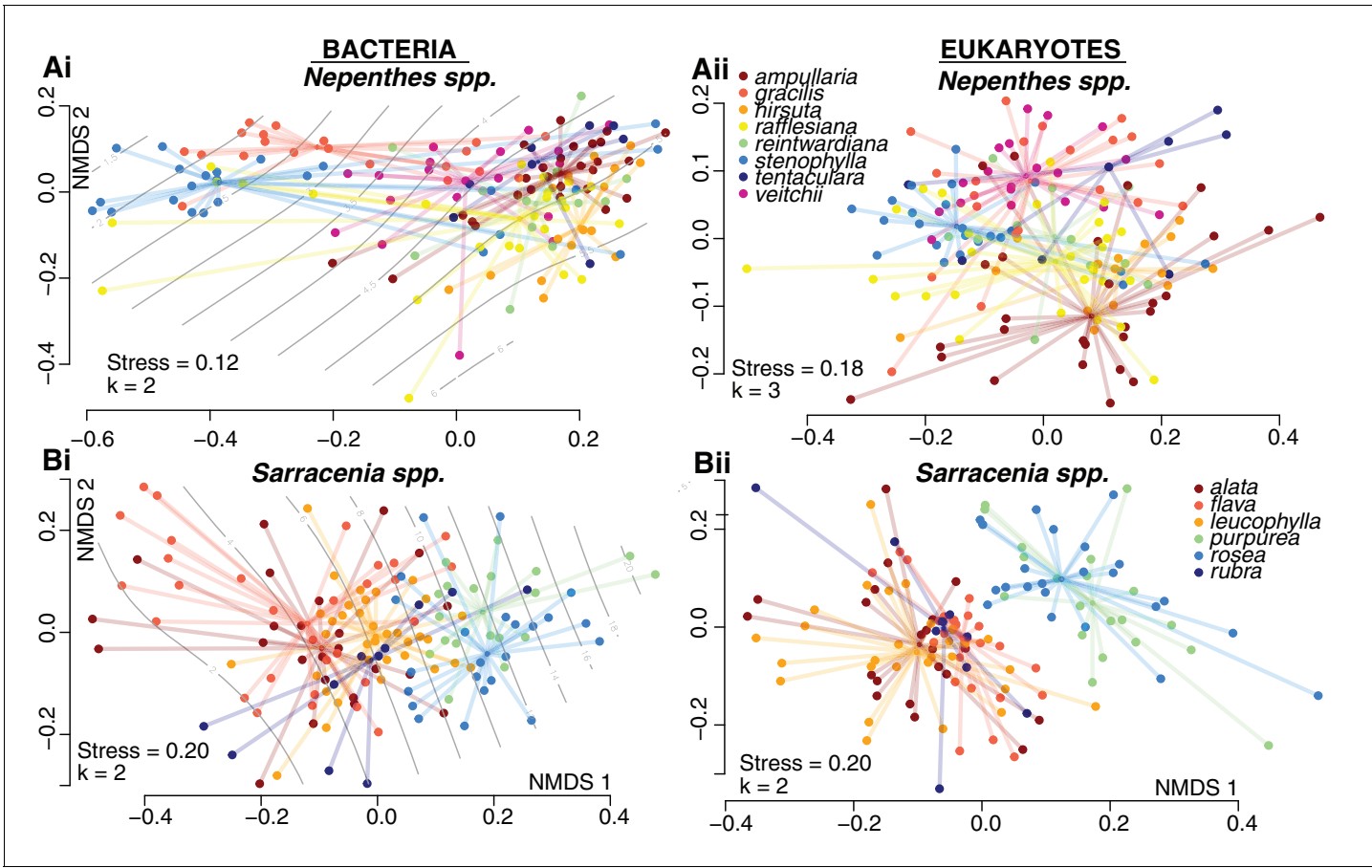

**Figure 3.** Community compositions differ by host species within both *Nepenthes* and *Sarracenia* pitchers. NMDS ordinations of pitcher samples, colored by host species. Ordisurf vectors with correlations greater than 0.3 are mapped onto the ordinations: pH in (Ai) and volume in (Bi).
DOI: https://doi.org/10.7554/eLife.36741.007

The following figure supplements are available for figure 3:

**Figure supplement 1.** Shannon diversities of bacterial communities from *Nepenthes* and *Sarracenia* pitchers have non-linear relationships with pH.
DOI: https://doi.org/10.7554/eLife.36741.008

**Figure supplement 2.** A few samples drive a weak correlation between the DNA concentration and Shannon diversity of *Sarracenia* pitcher bacterial communities.
DOI: https://doi.org/10.7554/eLife.36741.009

Pitchers of different species maintain different levels of acidity, although these differences are more pronounced in the genus *Nepenthes* than in the genus *Sarracenia*. Certain *Nepenthes* species can actively raise or lower the acidity of individual pitchers by pumping protons into or out of pitcher fluid (*An et al., 2001*; *Moran et al., 2010*). In our sampling of natural populations, we measured values below pH 4 in *N. rafflesiana*, *N. gracilis*, and *N. stenophylla*. But low-pH pitcher fluid does not seem to correlate with the *Nepenthes* phylogeny: low-pH species are in different clades, separated by species with higher average pH levels (*Meimberg and Heubl, 2006*). Furthermore, each species with low pH pitchers also had pitchers with higher pH levels. The large pH gradient across the *Nepenthes* fluids in our samples was strongly correlated with bacterial community composition, and explained most of the observed variation (*Figure 3A*. ordisurf $R^2$ = 0.74, p<0.001; *mantel* r = 0.63, p<0.001). This result supports a recent study that also noted a correlation between pitcher fluid pH and *Nepenthes* bacteria (*Kanokratana et al., 2016*). But the strong effect of pH on bacterial community composition is not driven by *Nepenthes* species differences per se; significant, high correlations between pH and bacterial community composition are also found within each of the three species with very low pH values when the data of each species are analyzed alone (Mantel tests: *N. gracilis* r = 0.46, p<0.001; *N. rafflesiana* r = 0.68, p<0.001; *N. stenophylla* r = 0.85, p<0.001). Eukaryotic community composition in *Nepenthes* was more weakly correlated with pH, and pH explained a smaller portion of the variation (*ordisurf* $R^2$ = 0.20, p<0.001; *mantel* r = 0.14, p<0.001). In the genus *Sarracenia*, bacterial (but not eukaryotic) community composition correlated with pH (bacteria: *ordisurf* $R^2$ = 0.15, p<0.001, *mantel* r = 0.11, p=0.010). For both the *Nepenthes* and *Sarracenia* bacterial communities, the relationship of pH with Shannon alpha diversity appeared to be quadratic: Shannon diversity peaked around pH 5.5 and was lower at both lower and higher pH levels (*Figure 3—figure supplement 1*). The correlation was much stronger for *Nepenthes* samples ($R^2$ = 0.67, p<0.001), but still significant for *Sarracenia* samples ($R^2$ = 0.07, p=0.002).

Shape emerges as a potential strong influence among the *Sarracenia* species, but is confounded with species identity: *S. purpurea* and *S. rosea* pitchers grow with a shorter, more cylindrical shape, while pitchers of *S. alata*, *S. flava*, *S. leucophylla* and *S. rubra* grow to a taller, more tapered shape. Although our samples of *S. purpurea* and *S. rosea* were collected in Massachusetts and Florida, respectively, the two species are very closely related (*Ellison et al., 2012*). The taller, tapered *Sarracenia* pitchers have an aspect ratio of width to height below 0.2; while the shorter, more cylindrical *Sarracenia* have an aspect ratio above 0.2, as do the *Nepenthes* pitchers from this study. Because growth form is confounded with *Sarracenia* host species identity and phylogeny, we could not analyze it as a separate variable. But pitchers from species of *Sarracenia* with shorter, wider pitchers tended to have a larger volume of fluid than the taller pitchers, and volume was strongly correlated with *Sarracenia* bacterial community composition (*ordisurf* $R^2$ = 0.31, p<0.001, *mantel* r = 0.15, p=0.006) and eukaryotic community composition (*ordisurf* $R^2$ = 0.18, p<0.001, *mantel* r = 0.17, p<0.001).

Collection site also significantly influenced *Sarracenia* communities; however, the effect was weaker when we controlled for the fact that not all species grow at all sites (*Supplementary file 1* Table S4). DNA concentration was significantly correlated with *Sarracenia* bacterial community composition (*ordisurf* $R^2$ = 0.22, p<0.001, *mantel* r = 0.11, p=0.035). There was also a weak, but marginally significant correlation between DNA concentration and the Shannon diversity of *Sarracenia* bacterial communities, which was driven by a few *Sarracenia* samples with very cloudy fluid and high relative abundances of Enterobacteriaceae OTUs (*Figure 3—figure supplement 2*).

## Relocated *Nepenthes* converge on *Sarracenia*-like communities

In a manipulative experiment, we relocated *Nepenthes* pitcher plants (propagated in Southeast Asia and purchased through a commercial U.S. nursery) to a *Sarracenia* bog in North America to test whether relocated *Nepenthes* pitchers would acquire communities similar in community structure and phylogenetic composition to those of local *Sarracenia*. All *Nepenthes* placed into the *Sarracenia* habitat were maintained in pots with soil material purchased in the U.S., and the plants were removed after experiments concluded. We included potted *S. purpurea* as a control to explore whether growth in a pot influenced community assembly. Target pitchers approaching maturity were manually opened in the bog on Day 1 of each experiment. The experiment also included cylindrical, round-bottomed 50 mL sterile glass tubes, either with or without sterilized insect material ('prey') added as a nutrient control (*Figure 4A*). During the experiment, we also recorded whether pitchers

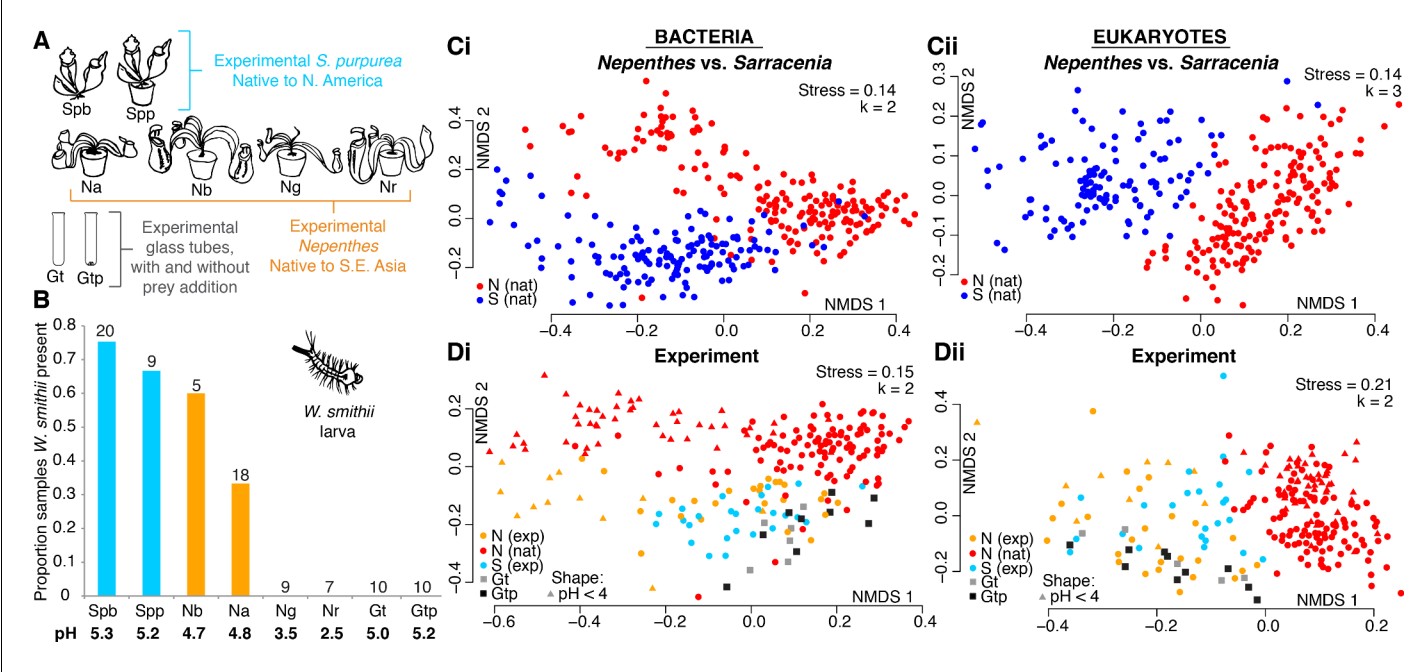

**Figure 4.** A manipulative field experiment demonstrates that *Nepenthes* pitchers in a *Sarracenia* habitat assemble *Sarracenia*-like microcosms. (A) Experimental treatments: Spb = *Sarracenia purpurea* bog; Spp = *S. purpurea* pot; Na = *Nepenthes ampullaria*; Nb = *N. bicalcarata*; Ng = *N. gracilis*; Nr = *N. rafflesiana*; Gt = glass tube; Gtp = glass tube with sterilized prey. (B) *Wyeomyia smithii* (pitcher plant mosquito) larvae colonized their native *S. purpurea* hosts, as well as foreign *Nepenthes* species with average pH > 4, but not pitcher-shaped glass tubes. Numbers of samples are listed above each category, and average pH values are listed below. (C) Natural microcosms of *Nepenthes* and *Sarracenia* pitchers sampled in SE Asia or North America house different organisms (Ci and Cii). (D) However, experimentally-relocated *Nepenthes* converge on *Sarracenia*-like communities and differ from those of natural *Nepenthes* (Di and Dii), except for bacterial communities sampled from *Nepenthes* in which pH < 4 (Di). Glass tubes (with or without added prey) of a pitcher-like form assemble communities that are similar to those of experimental pitchers (Di and Dii).
DOI: https://doi.org/10.7554/eLife.36741.010

contained larvae of the pitcher plant mosquito, *Wyeomyia smithii*, a specialized insect that completes its lifecycle only within *Sarracenia purpurea* pitchers (**Figure 4B**). *W. smithii* larvae regularly colonized their native *S. purpurea* pitchers (whether they were growing in the ground or in a pot). Surprisingly, they also colonized pitchers of *Nepenthes bicalcarata* and *N. ampullaria*, albeit in lower proportions (**Figure 4B**). The mosquitoes never colonized the more acidic *N. gracilis* and *N. rafflesiana* species, nor the experimental glass-tube pitchers.

To compare the biodiversity of entire communities, we first re-plotted our beta-diversity results from natural *Nepenthes* versus natural *Sarracenia* pitchers (**Figure 4C**), and found that community composition was significantly different for the two genera. Bacterial assemblages were more similar between the two genera than eukaryotic assemblages, and correspondingly, host genus explained less variation in bacterial than in eukaryotic community composition (Bacteria: *envfit* $R^2 = 0.33$, p<0.001, *adonis* $R^2 = 0.09$, p<0.001; Eukaryota: *envfit* $R^2 = 0.55$, p<0.001, *adonis* $R^2 = 0.14$, p<0.001).

We next compared the beta-diversity results of wild *Nepenthes* to our experimental data of relocated *Nepenthes*, native *Sarracenia*, and experimental glass tubes (**Figure 4D**). When *Nepenthes* microcosms assembled in a *Sarracenia* habitat, the assemblages of both bacteria and eukaryotes converged on compositions similar to those of the local *Sarracenia* and not wild *Nepenthes* (**Figure 4D**). The exception to convergence was in *Nepenthes* pitchers with pH below 4. The bacterial assemblages in highly acidic pitcher fluids (generally *N. gracilis* and *N. rafflesiana*) separated from other *Nepenthes* and *Sarracenia* pitchers in the same manner as acidic pitcher bacterial assemblages shifted in natural *Nepenthes* populations, and were instead more similar in structure and phylogenetic composition to wild, acidic *Nepenthes* (**Figure 4D** and **Figure 3**). Acidity explained most of the variation in bacterial community composition from the experiments (pH:

*ordisurf* R$^2$ = 0.67, p<0.001, *mantel* r = 0.50, p<0.001) and was also a significant predictor of eukaryotic community composition (pH: *ordisurf* R$^2$ = 0.21, p<0.001, *mantel* r = 0.11, p<0.001). Region—whether the pitchers were in Harvard Forest, Singapore or Malaysia—explained only a small portion of bacterial variation (*envfit* R$^2$ = 0.19, p<0.001, *adonis* R$^2$ = 0.07, p<0.001), but a larger portion of the eukaryotic variation (*envfit* R$^2$ = 0.41, p<0.001, *adonis* R$^2$ = 0.10, p<0.001).

Communities of bacteria and eukaryotes in our experimental pitchers were different from bog water communities (not shown), but partially clustered with the organisms colonizing the glass tube pitchers (*Figure 4D*). NMDS plots indicate glass tubes with added prey did not assemble communities more similar to the experimental pitcher communities than glass tubes without added prey. Only a very small portion of the variation in community composition was explained in analyses of pitchers vs. glass tubes; however, the differences were highly significant (bacteria: *envfit* R$^2$ = 0.05, p<0.001, *adonis* R$^2$ = 0.02, p<0.001; eukaryotes: *envfit* R$^2$ = 0.09, p<0.001, *adonis* R$^2$ = 0.03, p<0.001). The analysis suggests a sterile, pitcher-shaped form is almost, but not quite entirely, sufficient for acquiring a pitcher plant-like microcosm (*Figure 4D*).

## *Nepenthes* and *Sarracenia* microbiomes are enriched in gene pathways for degradation; both have high relative abundances of chitinases and key nitrogen mineralization enzymes

To investigate the functional potential of pitcher microbiomes, we generated metagenomes from 24 field-collected (not experimentally relocated) pitcher samples (16 *Nepenthes* and 8 *Sarracenia*). When compared with other published metagenomes for soil, lake, and phyllosphere samples (*Supplementary file 1* Table S5), pitcher plant community metagenomes were more enriched in gene pathways for fatty acid degradation, fermentation, and the biosynthesis of cell wall materials and non-proteinogenic amino acids, while non-pitcher metagenomes were more enriched in gene pathways for metabolic precursors (the biosynthesis of proteinogenic amino acids, tRNA charging, glycolysis, Calvin cycle and folate transformations) (*Figure 5A*).

In terms of overall functional potential as measured by Kegg Orthology (KO) groups using the Bray-Curtis dissimilarity metric in an NMDS plot, pitcher metagenomes were highly variable and were most similar to other phyllosphere communities (*Figure 5B*). Among *Nepenthes* and *Sarracenia* metagenomes, KO gene families in *Sarracenia purpurea* clustered close to *Nepenthes ampullaria*, *N. gracilis* and *N. reintwardiana*. Other *Sarracenia* (those with a tapered shape) and other *Nepenthes* (those with more acidic fluid) appeared to be more dissimilar in terms of functional potential.

To probe the functional similarities of *Nepenthes* and *Sarracenia* communities more deeply, we chose to compare abundances of enzymes involved in the degradation of complex polysaccharides and proteins. We specifically chose to focus on chitinases (K01183, GH families 18 and 19) because chitin is the main component of insect exoskeletons and can be used as both a carbon and nitrogen source. Because pitchers evolved to trap insect prey, we hypothesized pitcher plant microbiomes would have the genes to digest chitin. We also chose to focus on key enzymes involved in proteins and amino acid degradation (aminopeptidase N, lysine decarboxylase, ornithine decarboxylase, and glutamate dehydrogenase), because nitrogen mineralization via the microbiome may assist pitcher plants in nitrogen acquisition from prey. Finally, we chose to compare cellulases across the metagenomes, as we hypothesized microbiomes of pitcher plants would be less involved in breaking down plant material, compared to microbiomes of other habitats. Although individual pitcher samples showed considerable variability, *Nepenthes* and *Sarracenia* microbiome metagenomes did in fact have high relative abundances of chitinase, lysine decaboxylase, and ornithine decarboxylase genes compared to metagenomes of bacterial communities collected from other habitats (*Figure 5C*). But aminopeptidase N levels were not significantly higher in pitcher plants than in other habitats, and surprisingly, glutamate dehydrogenases were, in fact, significantly lower in pitcher plants. The glutamate dehydrogenase pathway can be either catabolic or anabolic, and this dual activity might explain our result. Cellulases were significantly lower in pitcher plant microbiomes, as hypothesized. Our results are consistent with recent proteomics data documenting high levels of amino acid and carbohydrate metabolism pathways in *Sarracenia pupurea* (*Northrop et al., 2017*). As the specific functions of the microbial communities within pitchers become better understood, we anticipate our data will be used to more explicitly measure and compare the functions of these different microbiomes.

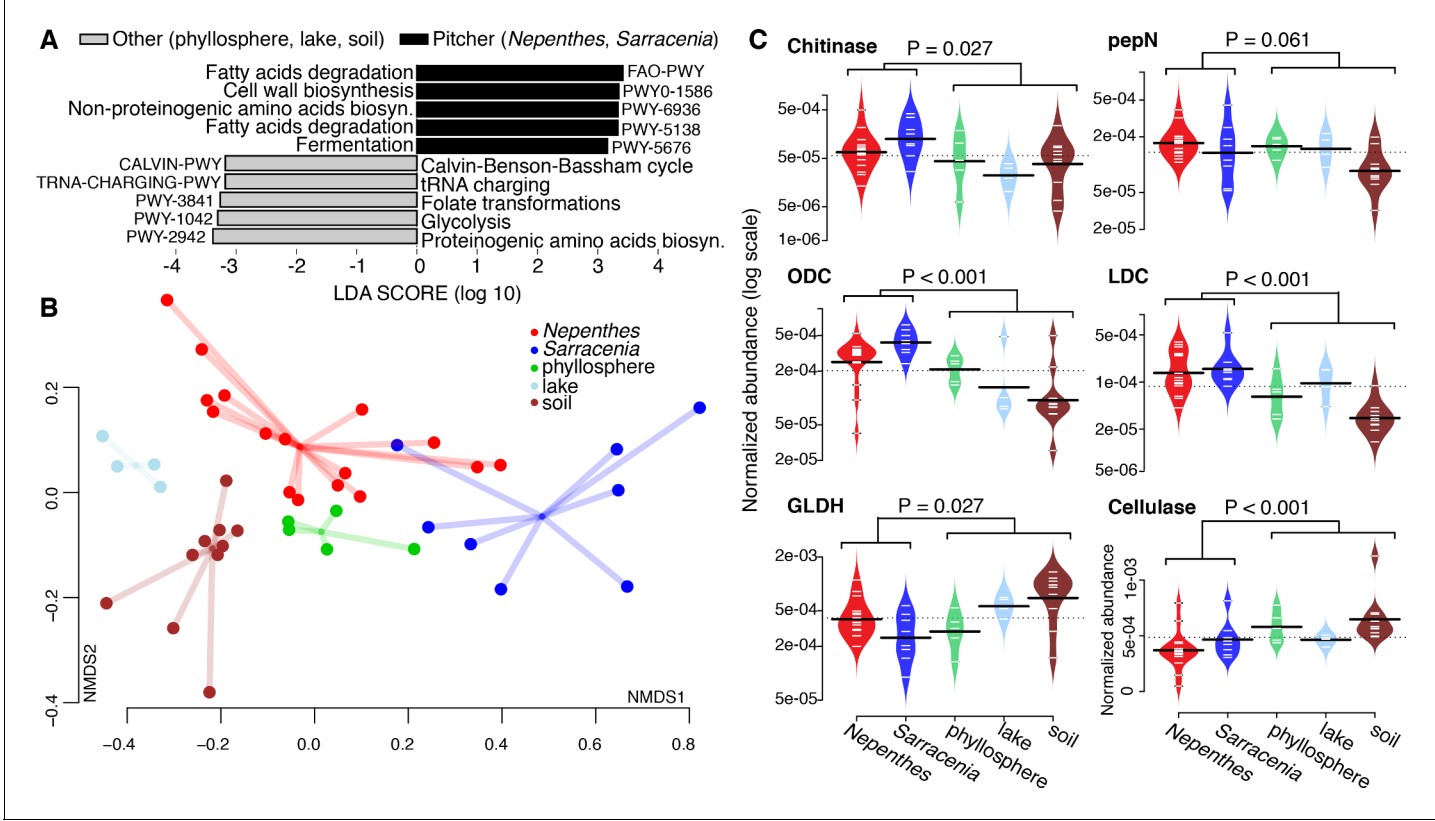

**Figure 5.** Pitcher plant microbiomes are enriched in degradation genes. (**A**) Gene pathways enriched in pitcher plant *versus* other environmental metagenomes. (**B**) NMDS plot of functional gene families comparing pitcher plant and environmental metagenomes. (**C**) Relative abundance of gene families in metagenomes. Abbreviations: pepN = aminopeptidase N; ODC = ornithine decarboxylase; LDC = lysine decarboxylase; and GLDH = glutamate dehydrogenase.

DOI: https://doi.org/10.7554/eLife.36741.011

## Discussion

Evidence for the convergence of communities within the carnivorous pitchers of *Nepenthes* and *Sarracenia* is strong and pitcher characteristics appear to regulate fundamental aspects of community biodiversity. First, in nature, the bacterial and eukaryotic communities inside both *Nepenthes* and *Sarracenia* pitchers are less species rich and less even than communities in surrounding soil or bog water; pitcher habitats favor a subset of available species (*Figures 1* and *2*). Second, although the communities within the two genera of pitchers are made up of different species, organisms tend to be closely related and from the same phylogenetic clades (*Figure 2*). This pattern was especially pronounced among bacteria. Third, both *Nepenthes* and *Sarracenia* pitcher microbiomes are depleted in pathways for metabolic precursors and enriched in pathways involved in fatty acid degradation and cell wall biosynthesis; pitcher microbiomes also possess a high relative abundance of both chitinases and genes involved in amino acid degradation (*Figure 5*). Results suggest the pitcher microbiomes of both genera function as decomposers of insect prey. Finally, *Nepenthes* pitchers experimentally placed into a *Sarracenia* habitat assembled *Sarracenia*-like communities (*Figure 4*). Convergently evolved pitchers appear to cause convergent interactions (*Bittleston et al., 2016b*) between the two genera and their associated pitcher microcosms.

Among species within a genus, fluid acidity was the strongest driver of beta-diversity, specifically for *Nepenthes* bacterial communities; this same characteristic explains aspects of our manipulative experiment. When microcosms of *Nepenthes* and *Sarracenia* pitchers assembled in parallel in a common environment, pitchers with similar fluid acidity held communities more alike in composition (*Figure 4*). Furthermore, while the communities formed in sterile glass tubes with or without sterilized insect 'prey' appeared somewhat similar to plant pitcher communities, communities were still

statistically different—suggesting that a general tube-like form that drowns insects in rainwater is almost, but not completely, sufficient for generating a pitcher-like microcosm. Other unmeasured characteristics of real *Nepenthes* and *Sarracenia* pitchers, including for example the production or abundance of plant-produced digestive enzymes or nectar, oxygen levels, and temperature, are also likely to cause these plant-formed pitcher microcosms to be more similar to each other than they are to glass tube microcosms.

The pitcher environment appears to be the dominant selective force shaping community composition in pitchers. Fewer eukaryotes consistently colonize *Nepenthes* or *Sarracenia* pitchers, as compared to bacteria (*Figure 2*); the pattern is consistent with stronger habitat filtering with increasing body size, as observed in bromeliad phytotelmata (*Farjalla et al., 2012*). Within each pitcher plant genus, collection site explained less of the observed variation than characteristics of the pitcher itself, suggesting that environmental filtering has a larger influence than dispersal. At a broad scale, comparisons between Southeast Asia and North America reveal that the regional pools of microbial organisms are distinct, and dispersal likely plays a stronger role in differentiating the microbial communities of these two continents. However, despite the differences in regional pools, organisms from the same phylogenetic groups colonize the pitcher microcosms found on opposite sides of the globe.

Convergent interactions, albeit on a much smaller scale, mirror the biome concept (*Odum, 1971*): the same functional groups of plants and animals are found in different regions of the world when those regions possess similar climate and soil conditions. For example, clumped grasses, crustose lichens, and jumping rodents are found in xeric shrublands globally, including in Australia and Arizona, just as Chitinophagaceae bacteria, Sphingomonadaceae bacteria, and Histiostomatidae mites are found in the unrelated pitcher microcosms of Southeast Asia and North America. However, beyond scale, there is a second fundamental difference between the two concepts: convergent interactions are, by definition, *interactions* among different living organisms, and thus there is a potential for reciprocal feedback, and potentially coevolution.

The concept of convergent interactions can be used to better understand the selective pressures structuring microbial biodiversity elsewhere, outside of the pitcher plant system. In addition to diet (*Godoy-Vitorino et al., 2012*; *Delsuc et al., 2014*; *Baldo et al., 2017*; *Sanders et al., 2017*; *Hu et al., 2018*), many other features of convergently evolved organisms appear to cause associations with similar microbial communities. For example, bacterial communities associated with fungus-growing ants, beetles, and termites in different regions of the world have dominant community members from the same genera, with convergent functional potential (*Aylward et al., 2014*), and microbial symbionts of sponges are functionally equivalent across phylogenetically divergent hosts (*Fan et al., 2012*). Convergent interactions can provide predictions about community structure and function, which can be tested across systems. The framework provides a tool to explore compositional and functional similarities of whole ecosystems, potentially enabling an understanding of the fundamental evolutionary and environmental drivers structuring microbial communities.

## Materials and methods

**Key resources table**

| Reagent type (species) or resource | Designation | Source or reference | Identifiers | Additional information |
|---|---|---|---|---|
| Commercial assay or kit | MoBio PowerClean kit | Qiagen/MoBio | | |
| Commercial assay or kit | Quant-iT High-Sensitivity dsDNA Assay Kit | Invitrogen/ThermoFisher | | |
| Commercial assay or kit | TruSeq DNA PCR Free HT Kit | Illumina | | |
| Commercial assay or kit | KAPA LTP Library Prep Kit | Roche | | |
| Commercial assay or kit | KAPA Library Quantification Kit | Roche | | |
| Commercial assay or kit | PerfeCta NGS Library Quantification Kit | Quanta Biosciences/VWR | | |

Collections and experiments took place from 2012 to 2014 at field sites along the U.S. Gulf Coast, at Harvard Forest in Massachusetts, in Singapore, and in the Maliau Basin of Borneo (*Figure 1A*). In total, we sampled and sequenced communities collected in the field from more than 330 pitchers from 8 species of *Nepenthes* and 6 species of *Sarracenia* (for more details about the species, see

*Supplementary file 1* Table S1), and 70 environmental samples. The experimental data included samples from 60 experimental pitchers of natural and potted *S. purpurea* and 4 species of potted *Nepenthes*, and 16 glass tubes. R code and data for our analyses are available via the Harvard Dataverse: https://dataverse.harvard.edu/dataset.xhtml?persistentId=doi:10.7910/DVN/QYUBN2.

## Field collections and background information

*Nepenthes* pitchers from three co-occurring species (*N. ampullaria*, *N. gracilis* and *N. rafflesiana*) were sampled from three sites in Singapore (Kent Ridge Park, Bukit Timah Nature Preserve, and between Lower and Upper Peirce Reservoir Park) in January 2012. Additional pitchers from the same species and sites were sampled in March 2013 and March 2014. Pitchers from an additional five co-occurring species (*N. veitchii*, *N. tentaculata*, *N. stenophylla*, *N. reinwardtiana*, and *N. hirsuta*) were sampled from the Maliau Basin, Borneo in March 2014.

Sarracenia pitchers from five species (*S. alata*, *S. flava*, *S. leucophylla*, *S. rosea* and *S. rubra*) were sampled from thirteen sites from Mississippi to Florida along the U.S. Gulf Coast in June 2014 and a sixth species (*S. purpurea*) was sampled from Harvard Forest in Massachusetts in July 2014. For details of which species were sampled from which sites see *Supplementary file 1* Table S1. Sites were considered different if separated by more than 0.1 degree of latitude or longitude.

## Sampling pitcher fluid

Contents of each pitcher were collected with sterile, single-use plastic transfer pipettes and placed into empty, sterile plastic tubes. Fluids were mixed within each pitcher using the pipette before collecting to homogenize any differences by depth. Volumes and pH levels of all pitcher fluids were recorded, except for our first collection in Singapore in 2012 (for more detail on Singapore sampling see [*Bittleston et al., 2016a*]). The pitchers of some species can have large volumes (e.g. 100–500 mLs); for higher volume samples, we estimated total volume and collected a well-mixed subsample from the pitcher. We measured pH with colorpHast strips (EMD Millipore) by removing small amounts of fluid with additional sterile pipettes. To preserve DNA, we added cetyltrimethylammonium bromide and salt solution (hereafter 'CTAB'; final concentrations: 2% CTAB, 1.4 M NaCl, 20 mM EDTA, 100 mM Tris pH 8) to each sample in the same volume as the collected fluid. All samples were processed the same day as collection, except for Maliau Basin samples that were refrigerated overnight and processed the next morning, due to time constraints. After CTAB addition, samples were transported at room temperature to Harvard University, and subsequently frozen.

## Sampling the surrounding environment

Protocols reflected pitcher plant habitats (*Figure 1A*). When wet, we collected bog samples from the surrounding environment, and when dry, we collected soil samples and water either from fallen leaves or from sterile tubes placed in the environment, as follows: Singapore, March 2013—soil, Gulf Coast, June 2014—soil and bog water; Massachusetts, July 2014—bog water; Singapore, February 2014—sampling from plastic tubes left out for one month to collect rainwater and acquire microbial communities; Maliau Basin, January 2014—sampling from soil and water held in fallen leaves. All soil samples were collected from the surface organic layer in approximately 7 mL volumes. See Dataset S1 for sample details.

## Experimental relocation of *Nepenthes spp.* to a New England bog

In summer 2013 we set up experiments to manipulate *Nepenthes* within a *Sarracenia* habitat, the Tom Swamp bog at Harvard Forest in Petersham, MA (USA). Four different species of *Nepenthes* (*N. ampullaria*, *N. bicalcarata*, *N. gracilis* and *N. rafflesiana*) were purchased from Borneo Exotics via the ExoticPlantsPlus nursery in New York. *Nepenthes* plants were maintained in a greenhouse for two months after arriving from Southeast Asia, and then a growth chamber for a few weeks while pitchers were maturing for use in the field. *Sarracenia purpurea* plants were purchased from Meadowview Biological Research Station in Virginia, potted using purchased sphagnum peat and perlite, maintained in a greenhouse for three months, and used in the experiments as a control for whether growth in a pot influenced community assembly.

Experiment I had six treatments: *S. purpurea* growing naturally in the bog, *S. purpurea* in pots, *N. ampullaria*, *N. gracilis* and *N. rafflesiana* in pots, and empty, 50 mL sterile glass tubes used as a

rough mimic of the pitcher shape. There were eight stations, and the experiment ran from June 26 – July 17. Pitchers were sampled for subsequent sequencing on days 14 and 21. Experiment II had five treatments: *S. purpurea* in the bog, *S. purpurea* in pots, *N. ampullaria* in pots, sterile glass tubes, and sterile glass tubes each filled with 30 mg of autoclaved, ground wasps as a nutrient and prey control. There were five stations, and the experiment ran from July 17 – September 4. Pitchers were sampled for subsequent sequencing on days 14, 35, and 49. Experiment III had six treatments: *S. purpurea* in the bog, *N. ampullaria*, *N. rafflesiana* and *N. bicalcarata* in pots, sterile glass tubes, and sterile glass tubes with prey. There were five stations, and the experiment ran from July 24 – September 10. Pitchers were sampled for subsequent sequencing on days 15, 35, and 48.

To sample, we collected 750 uL of fluid from experimental pitchers and tubes using sterile transfer pipettes, as described above. At sampling, we also noted the presence or absence of pitcher plant mosquito larvae (*Wyeomyia smithii*) in pitchers. On the last day of each experiment, we collected entire pitcher contents. On the last days of Experiments II and III, we collected samples of bog water. All samples were stored in small tubes in a cooler with ice, and brought back to the laboratory, where they were frozen the same day. Subsequent analyses target only the last day's sample from each pitcher or tube, so that no pitcher or tube is included more than once in the dataset.

## Sample processing, DNA extraction, amplification and sequencing

Once we had collected samples, we turned our attention to DNA extraction and sequencing. When removing fluid for DNA extraction, we took care to avoid macroscopic organisms. We concentrated sample fluid by filtration or isopropanol precipitation and centrifugation. Concentration protocol did not affect community composition (samples processed by different techniques clustered together).

We extracted DNA by bead-beating concentrated materials with buffer and phenol-chloroform, and then proceeding with a standard phenol-chloroform extraction (*Sambrook and Russell, 2001*). We included a negative control with each set of extractions, and discarded the samples from one extraction set found to have measurable amounts of DNA in the negative control. We measured DNA quantity, and then re-extracted DNA from a few samples with very low DNA amounts, using a larger initial volume. DNA extracts with dark coloration (suggesting high levels of polyphenols) were cleaned using a MoBio Powerclean kit. DNA concentrations of successful extractions were fluorometrically quantified a final time using a Quant-iT High-Sensitivity dsDNA Assay Kit (Invitrogen). Samples were sent to Argonne National Laboratories for Illumina MiSeq next-generation amplicon sequencing. The Earth Microbiome Project's barcoded 16S and 18S primers (*Amaral-Zettler et al., 2009*; *Caporaso et al., 2012*) were used to amplify DNA in separate runs, with PCR amplification and sequencing executed according to the Earth Microbiome Project protocols (http://www.earthmicrobiome.org/emp-standard-protocols). The 16S primers target the V4 region of the ribosomal RNA gene and are used to characterize prokaryotic communities, while the 18S primers target the V9 region and are used to characterize eukaryotes. The amplicon sequencing datasets can be accessed from the Sequence Read Archive as NCBI BioProject PRJNA448553.

## Shotgun metagenomics

To explore functional gene diversity in pitcher plant microbiomes, we conducted shotgun metagenomic sequencing with 24 of our pitcher samples. Sixteen samples were sequenced at the High Impact Research Institute in Malaysia (HIR) at the University of Malaya, and eight at the Bauer Core Facility (RRID:SCR:001031) at Harvard University. The HIR set targeted two samples from each of four different species of both *Nepenthes* and *Sarracenia*. DNA was extracted as described above, but 2–4 extractions were done for each sample and resulting DNA was pooled to increase the amount available for sequencing. DNA was sheared with a Covaris at settings aiming for average lengths of 350 base pairs (bp), and libraries were prepared using a TruSeq DNA PCR Free HT Kit. DNA library concentrations were measured using a KAPA Library Quantification Kit, and the qualities were tested with a Bioanalyzer. DNA libraries were then pooled in equal concentrations and sequenced on the Illumina HiSeq 2500 platform in four Rapid Run, paired-end,100 bp lanes.

The Bauer Core Facility set included eight additional *Nepenthes* samples (two samples from each of four different species). Here, we used the same DNA extractions as previously used for metabarcoding. DNA samples were sheared with a Covaris at 500 bp, and prepared with a KAPA LTP Library Prep Kit. Due to lower initial DNA quantities, the samples were subject to 2–9 cycles of PCR before

final quantification using a PerfeCta NGS Library Quantification Kit, quality testing with a Bioanalyzer, and pooling of samples in equal concentrations. These libraries were sequenced in one-third of an Illumina HiSeq 150 bp paired-end Rapid Run lane. Shotgun metagenomic data can be accessed via the Argonne National Laboratory metagenomics server MG-RAST (RRID:SCR:004814): http://www.mg-rast.org/linkin.cgi?project=mgp15454.

## Analyses of 16S and 18S diversity

To generate Operational Taxonomic Units (OTUs), amplicon data were clustered using QIIME (Quantitative Insights Into Microbial Ecology, RRID:SCR:001905) versions 1.8 and 1.9 (*Caporaso et al., 2010*) on Harvard University's Odyssey computing cluster. We joined forward and reverse reads using fastq-join, then split libraries with a PHRED quality cut-off of 20 to remove low-quality sequences, and used UCLUST (version 1.2.21q) open-reference clustering to form groups of sequences into OTUs with 97% similarity. Resulting numbers of sequences and OTUs are summarized in *Supplementary file 1* Table S2. Phylogenetic trees were generated using QIIME default settings for 16S; when generating the18S alignment and tree we set the allowed gap fraction to 0.8 and the entropy threshold to 0.0005. We assigned taxonomy with the greengenes version 13_8 (Greengenes Database Consortium) and SILVA version 111 databases for 16S and 18S, respectively. For 18S, we used the BLAST method to assign taxonomy, as UCLUST assignment was poor. For subsequent analyses of 18S sequences, we used only OTUs assigned to Eukaryota.

To calculate alpha diversity of our samples, we first discarded any samples with fewer than 6500 or 4300 sequences for the 16S and 18S datasets, respectively. We then built rarefaction plots of soil, bog, *Nepenthes*, and *Sarracenia* samples using the observed species metric in QIIME and standard deviations across each category (*Figure 1B*). We also calculated the Shannon diversity index and standard deviations across the same categories (*Figure 1B*, *Supplementary file 1* Table S3). To ensure that sample volume was not driving differences in alpha diversity between environmental and pitcher plant samples, we used a linear model to test for a correlation between sample volume and observed OTUs. We also re-calculated Shannon diversity and standard deviations using a subset of 155 samples where DNA was extracted from the same volume for all samples (*Supplementary file 1* Table S3).

To explore beta diversity among samples, we first removed any observation of an OTU with less than 10 sequences per sample in order to minimize the probability of including sequencing errors, including barcode misassignments (*Bokulich et al., 2013*; *Nelson et al., 2014*). We accounted for uneven sequencing across the samples by subsampling our OTU tables to 4000 sequences per sample. We calculated dissimilarity matrices with the unweighted Unifrac metric (*Lozupone and Knight, 2005*) and Bray-Curtis using R packages *picante* and *phyloseq* (*Kembel et al., 2010*; *McMurdie and Holmes, 2013*; RRID:SCR:013080), and ran non-metric multidimensional scaling (NMDS) analyses using the *vegan* R package (*Oksanen et al., 2013*; RRID:SCR:011950) (*Figures 1C*, *3*, *4* and *5* and *Figure S1*). We used the functions *envfit* and *ordisurf* to fit environmental factors or vectors (respectively) to our ordinations and to analyze main effects. We also calculated dissimilarity matrices using the Bray-Curtis metric, to take into account how relative abundances among OTUs might influence beta-diversity analyses. For both the unweighted Unifrac and Bray-Curtis measures, we used mantel tests to test for correlations of the dissimilarity matrices with pH and volume, and permutational multivariate analyses of variance (PERMANOVAs, function *adonis* in *vegan*) (*Anderson, 2001*) to test the explanatory power of factors including plant species and collection site (*Supplementary file 1* Table S4). We adjusted P-values within each group to account for multiple comparisons using the Benjamini-Hochberg procedure (*Benjamini and Hochberg, 1995*).

To examine phylogenetic patterns among *Nepenthes* pitchers, *Sarracenia* pitchers, and environmental samples, we chose to focus on relatively common OTUs, removing OTUs containing fewer than 100 sequences across all our samples as well as those not present in at least 10% of either *Nepenthes* or *Sarracenia* microbiome samples. We then subsampled the OTU table for each category to 2000 sequences per sample, combined all observations of the OTUs by category (e.g. *Nepenthes*, *Sarracenia* or environment), and normalized by the number of samples in each category. We filtered our previously generated 16S and 18S phylogenetic trees using the resulting OTU tables and plotted them with the Interactive Tree of Life (iToL) program(*Letunic and Bork, 2011*) (*Figure 2*). The bacterial tree was rooted with Archaea, and the eukaryotic tree was rooted in Streptophyta (land plants and most green algae). We added barcharts along the outer edge of the trees,

displaying the natural log of the abundance for each OTU in each category, and gray dots to each branch with bootstrap support of 0.7 or higher (*Figure 2*). The same trees, with branch lengths and tree scales included, are shown in *Figure 2—figure supplement 1*. The branches of the figure supplement trees were colored by either phylum (for bacteria) or by broad taxon levels (for eukaryotes).

## Functional analyses

For the shotgun metagenomic data, we combined forward and reverse reads from all lanes for each sample, and used Trimmomatic to remove barcodes and low-quality sequences. We used HUMAnN2 (HMP Unified Metabolic Analysis Network 2, [*Abubucker et al., 2012*]) on Harvard University's Odyssey computing cluster to identify individual reads by comparing and annotating reads to reads of known function, build profiles of identified functional genes for each sample, and normalize numbers of sequences across samples. We next compared our metagenomes to publicly available metagenomes from soil, lake, and phyllosphere habitats using MG-RAST (*Glass et al., 2010*; RRID:SCR: 004814) and the NCBI's Sequence Read Archive (SRA; RRID:SCR:004891; accession numbers are listed in *Supplementary file 1* Table S5). We analyzed these metagenomes in the same way as our own data, also using HUMAnN2. To detect differentially abundant gene pathways in pitcher plants vs. the other metagenomes, we subset gene pathways to those with abundances and variances in the top 50% of the dataset, and used LEfSe (Linear Discriminant Analysis Effect Size, *Segata et al., 2011*) to identify statistically significant features. We reported the pathways with the five largest linear discriminant values for each group (*Figure 5*). We made NMDS plots of KO functional matrices with Bray-Curtis distances and a beanplot of chitinases using the *vegan* and *beanplot* (*Kampstra, 2008*) packages in R. We tested for differences in gene family abundances between pitcher plant and comparison metagenomes using Mann-Whitney U tests (function *wilcox.test* in R) and we adjusted P-values to control for false discoveries using the Benjamini-Hochberg procedure (*Benjamini and Hochberg, 1995*) (*Figure 5*).

## Acknowledgements

Collecting permits were issued to LSB by Singapore National Parks (NP/RP11-096), the Sabah Biodiversity Centre (Collecting permit: JKM/MBS.1000-2/2 (112); Export permit: JKM/MBS.1000-2/3 (97)), and the Maliau Basin Management Committee (MBMC 2012/10). Permission to collect in North America was granted by Harvard Forest, the Nature Conservancy, Eglin Air Force Base, Nokuse Plantation, and local National and State Forests. We are grateful for collecting help from Dan Sternof Beyer and Kadeem Gilbert, and for expedition support and use of lab space from the Institute for Tropical Biology and Conservation at the University of Malaysia Sabah. Thanks to Drs. Li Daiquin and Gregory Jedd for use of lab space in Singapore, and to Dr. Shawn Lum, Dr. Stuart Davies, Kang Min Ngo, and Su Shiyu for help locating *Nepenthes*. We deeply appreciate Dr. Tom Miller's invaluable suggestions and field support during Florida collecting trips, and the three reviewers whose helpful comments improved our manuscript.

## Additional information

### Funding

| Funder | Grant reference number | Author |
| --- | --- | --- |
| National Science Foundation | NSF Graduate Fellowship and Doctoral Dissertation Improvement Grant DEB-1400982 | Leonora S Bittleston |
| John Templeton Foundation | Foundational Questions In Evolutionary Biology | Naomi E Pierce Anne Pringle |
| Harvard University Museum of Comparative Zoology | Putnam Expedition Grant | Leonora S Bittleston |
| National Geographic Society | | Naomi E Pierce |

| | | |
|---|---|---|
| Universiti Malaya | High Impact Research Grant, UM-MOHE HIR Grant UM.C/625/1/HIR/MOHE/CHAN/14/1 | Kok Gan Chan |
| National Science Foundation | SES-0750480 | Naomi E Pierce |
| Universiti Malaya | High Impact Research Grant, H-50001-A000027 | Kok Gan Chan |
| Universiti Malaya | High Impact Research Grant, A-000001-50001 | Kok Gan Chan |

The funders had no role in study design, data collection and interpretation, or the decision to submit the work for publication.

## Author contributions

Leonora S Bittleston, Conceptualization, Data curation, Formal analysis, Funding acquisition, Investigation, Visualization, Methodology, Writing—original draft; Charles J Wolock, Investigation, Methodology, Writing—review and editing; Bakhtiar E Yahya, Resources, Supervision, Writing—review and editing; Xin Yue Chan, Resources, Data curation, Methodology, Writing—review and editing; Kok Gan Chan, Resources, Supervision, Funding acquisition, Methodology, Writing—review and editing; Naomi E Pierce, Conceptualization, Resources, Supervision, Funding acquisition, Methodology, Writing—original draft, Project administration, Writing—review and editing; Anne Pringle, Conceptualization, Resources, Formal analysis, Supervision, Funding acquisition, Visualization, Methodology, Writing—original draft, Project administration, Writing—review and editing

## Author ORCIDs

Leonora S Bittleston http://orcid.org/0000-0003-4007-5405
Anne Pringle http://orcid.org/0000-0002-1526-6739

## Decision letter and Author response

Decision letter https://doi.org/10.7554/eLife.36741.022
Author response https://doi.org/10.7554/eLife.36741.023

# Additional files

## Supplementary files

• Supplementary file 1. Tables S1-S5
DOI: https://doi.org/10.7554/eLife.36741.012

• Supplementary file 2. Sample Metadata
DOI: https://doi.org/10.7554/eLife.36741.013

• Transparent reporting form
DOI: https://doi.org/10.7554/eLife.36741.014

## Data availability

Amplicon sequencing data have been deposited as NCBI BioProject PRJNA448553: https://www.ncbi.nlm.nih.gov/bioproject/PRJNA448553. Metagenomic sequencing data have been deposited in MG-RAST: http://www.mg-rast.org/linkin.cgi?project=mgp15454. The source code and data for Figures 1-5 and for Tables S3 and S4 have been deposited in a Harvard Dataverse repository: https://doi.org/10.7910/DVN/QYUBN2.

The following datasets were generated:

| Author(s) | Year | Dataset title | Dataset URL | Database, license, and accessibility information |
|---|---|---|---|---|
| Bittleston L | 2018 | Convergence between the microcosms of Southeast Asian and North American pitcher plants | https://www.ncbi.nlm.nih.gov/bioproject/?term=PRJNA448553 | Publicly available at NCBI BioProject (accession no. PRJNA448553) |

| Bittleston L | 2018 | Pitcher plant shotgun metagenomics | http://www.mg-rast.org/linkin.cgi?project=mgp15454 | Publicly available at MG-RAST (accession no. mgp15454) |
| Leonora S Bittleston | 2018 | Data and code for: Convergence between the microcosms of Southeast Asian and North American pitcher plants | https://doi.org/10.7910/DVN/QYUBN2 | Publicly available at the Harvard Dataverse website (DOI: 10.7910/DVN/QYUBN2) |

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
