## [Decision Letter]

Thank you for submitting your article "Convergence between the microcosms of Southeast Asian and North American pitcher plants" for consideration by *eLife*. Your article has been reviewed by three peer reviewers, including Dianne Newman as Reviewing Editor, and the evaluation has been overseen by Detlef Weigel as Senior Editor. The reviewers have opted to remain anonymous.

The reviewers have discussed their reviews with one another and the Reviewing Editor has drafted this decision to help you prepare a revised submission.

Summary:

Though we find many aspects of your work to be positive, there are mixed opinions regarding whether it sufficiently advances the field.

Essential revisions:

We would like to give you an opportunity to respond to the reviewers' concerns (paying particular attention to the issue of how sample volume was accounted for in your estimations of OTU curves, and providing metadata that would allow for more meaningful interpretations). We also encourage you to revise the Introduction/Discussion to more pointedly articulate how this study compares to previous ones that examine microbial community composition (including in animals), and how it raises new questions that move the field forward conceptually.

*Reviewer #1:*

See summary comments

*Reviewer #2:*

This paper presents a compelling combination of observational data and experiments to test the hypothesis that distantly-related families of pitcher plants share a common core microbiota independent of their geographic locations and evolutionary histories. The authors first present data supporting the hypothesis that the inquiline microbial communities of *Nepenthes* and *Sarracenia* species are more similar to one another than they are to their external environments. Such similarity across these huge geographic distances suggests that the plants' leaf environments selectively enrich a similar suite of inquiline taxa and metabolic pathways. The authors then conduct a clever field experiment wherein *Nepenthes* host plants, which evolved in Asia, are brought into a *Sarracenia* habitat in New England and are allowed to naturally develop their microbiota. The authors found that *Nepenthes* pitcher community composition matched that of local *Sarracenia* pitchers, suggesting 1) that some shared features of these two families deterministically influenced the success of particular clades of micro-organisms, and 2) that differences between pitcher families' microbiota are not evolutionarily conserved, and are instead products of their local source pools. In terms of novelty, a number of other studies have previously documented convergent interactions among hosts and their microbial symbionts (e.g., N-fixing and mycorrhizal associations in plants, diet-driven gut microbiota in animals), but this is among the first to identify shared communities that encompass such a wide taxonomic breadth (arthropods, micro-eukaryotes, bacteria). It is also one of the first to experimentally test how these communities form when placed well outside of their natural ranges (though analogous 'natural experiments' have been conducted on *Sarracenia* introduced to Switzerland by Fragnière 2012 and Zander et al., 2016, which might be discussed). The inclusion of metagenomic data is another valuable contribution. Few studies have done so, and none as thoroughly as this one.

I outline a few concerns with the current draft below.

One major issue with the study design is that it appears pitcher leaves within each species were sampled at a single point in time from their respective locations. Given that pitcher environments and microbial communities have been observed to change markedly in response to season and prey deposition (e.g., Koopman et al., 2010; Sirota et al., 2013 PNAS) it would be very helpful to also have data on prey biomass (mg prey/ml), oxygen concentration, and (most importantly) living microbial and arthropod biomass in each pitcher. Were these data collected? How might differences in prey biomass influence the results?

Introduction, second paragraph: This paragraph draws some parallels between microbial diversity, global change, convergent interactions that are a bit of a stretch and are not covered in the design/Results/Discussion. If the goal is to justify the study, I think it's better to scrap the narratives of microbial diversity and global change, and instead introduce the question at hand – how microbial communities assemble in highly-specialized, convergently-evolved structures. This could be done by introducing the concept of convergence in dietary structures among various clades (e.g. ruminants and the hoatzin, blood-feeding arthropods, wood-eating insects) and then introducing the potential importance of microbial digestive associates in these cases. The case can then be made that pitcher plants are uniquely situated for testing this concept.

Introduction, third paragraph: Citations here are misleading. Why not cite the studies that have actually used pitcher plants to investigate metacommunities and community assembly? (e.g.., Kneitel and Miller 2003; Buckley et al., 2003; Armitage 2017).

Introduction, third paragraph: It's not clear what these 'clear parallels' are between these different communities. Here again, citations are somewhat misleading in that they do not address the aforementioned parallels between pitcher plant and phyllosphere/nectar/rhizosphere microbionts.

Introduction, last paragraph: Please explain how 'convergence' is defined. As it stands, it is a nebulous enough term that it can be used to assign meaning to a huge variety of different patterns, such as:

– many shared OTUs

– some shared numerically-abundant OTUs

– no shared OTUs but some shared genes conferring an implied function

– no shared OTUs but many shared genes conferring no implied function

– shared relative abundance of OTU families but no shared OTUs

– shared network of OTU co-occurrences among hosts

Are there any commonly-used definitions? Is there a criterion for rejecting convergence?

Also, if the authors are truly searching for evidence of convergent interactions, then why aren't interactions being measured? The authors are making the tacit assumption that shared communities equates to shared interactions within those communities, which is not necessarily the case.

Subsection “Relocated *Nepenthes* converge on *Sarracenia*-like communities”, last paragraph: Were pitcher leaves more similar to tubes with prey than they were to tubes without prey? These comparisons could identify how much variation prey capture accounts for, independent of the pitcher leaf.

Subsection “Relocated *Nepenthes* converge on *Sarracenia*-like communities”, last paragraph: Given the differences in sample sizes, it's entirely conceivable that more tube samples would recapitulate the observed variation among pitchers and the 'reduced variation' would vanish. Therefore, I suggest removing the statement that the glass communities were different from the experimental pitchers owing to their variation. Also, I couldn't find the PERMANOVA results for experimental pitchers vs. glass tubes. The negative results should also be included for the sake of transparency.

Discussion, second and fifth paragraphs: Without data on the actual contribution of microbes to host functioning (e.g., via prey decomposition/isotope uptake paired with fluid dilution/antibiotic treatment), calling these 'convergent interactions' remains speculative, as the term 'interaction' implies some type of reciprocal responses between the host organism and its microbiota. However, we already know that removing of many components of the pitcher food web (such as micro-arthropods) carries no perceptible consequences for their hosts (Butler, Gotelli and Ellison et al., 2008; Baiser, Ardeshiri and Ellison, et al. 2011). Again, a more formal introduction to 'convergent interactions' would be welcome.

Subsection “Experimental relocation of *Nepenthes spp.* to a New England bog”, second paragraph: I'm not convinced that the control groups (bog water, soil) are appropriate references to use for detecting convergence among pitchers. For instance, bog and soil water are treated as "null" environments to which pitcher leaves are compared. However, since pitcher leaves are really being colonised by microbes derived primarily from the guts and bodies of trapped or resident arthropods, then wouldn't those organisms' combined microbiota serve as a more meaningful source pool? That way, the resulting pitcher communities could be directly related to environmental filtering from the pool of colonists. What if the insect prey microbiota ended up being identical to the pitcher microbiota?

Subsection “Analyses of 16S and 18S diversity”, third paragraph: Unweighted UniFrac is notoriously sensitive to rare members of the community. Please also include similar weighted UniFrac or Bray-Curtis based dissimilarity analysis in the supplement so that readers can assess the influence of rare vs. dominant taxa on beta-diversity.

Subsection “Analyses of 16S and 18S diversity”, last paragraph: How were these trees generated? I can't assess their quality without more information such as the identification of major clades on the trees, some branch length scale, bootstrap values, rooting methodology, etc.

*Reviewer #3:*

This study compared the eukaryotic and prokaryotic inhabitants of two phylogenetically and geographically distinct clades of pitcher plants. The authors ask if these inhabiting organisms show convergence in taxonomic and functional composition, defined here as being more similar to each other in composition than to other neighbouring habitats.

Overall, this is a very pleasing study. The breadth of organismal diversity studied – from bacteria to insects – is impressive and made possible by barcoding and metagenomics methods. The authors combine observational evidence (pitcher communities in native country) with experimental evidence (transplant of one pitcher genera to the other country) to build a case for convergence mediated by habitat filtering. Lastly, the authors delve into mechanistic details, examining the role of water chemistry (pH was key) and characterizing the prevalence of functional pathways.

1) My most substantive comment is that I cannot find evidence in the Materials and methods or supplementary material that the authors standardized sample volume between the pitchers and the other environmental samples (soil and bog water). Since the authors use species accumulation curves (Figure 1B) to suggest that pitchers are more species depauperate than surrounding soil or bog water, standardizing sample volume is critical. Specifically, if there is any micro-scale spatial turnover in species composition, collecting a large sample, homogenizing it, and extracting 4000 sequences will result in capture of more OTUs that collecting a smaller sample, homogenizing it and extracting 4000 sequences.

2) In the analysis of within-genera correlates of community structure (subsection “Within each genus, pitcher species, acidity, form, and volume correlate with community composition”, third paragraph), it is not clear if you have controlled for species. If not, any community correlations with the physical and chemical properties of pitchers would be psuedoreplicated as these physical/chemical differences could simply stand in for species differences.

3) In the gene pathways analysis, I couldn't determine if you were using samples from the same region (transplant experiment) or different regions (observation dataset). This matters in the interpretation of the results.

4) There has been a fair amount of work comparing the strength of habitat filtering on macroscopic organisms vs. microbial, for example in lakes and bromeliads. Your study seems well positioned to follow up these ideas, in particular Figure 2 suggests that the pitcher habitat filters bacteria and eukaryotes to different degrees. Can you incorporate in your Discussion?

---

## [Author Response]

Essential revisions:We would like to give you an opportunity to respond to the reviewers' concerns (paying particular attention to the issue of how sample volume was accounted for in your estimations of OTU curves, and providing metadata that would allow for more meaningful interpretations). We also encourage you to revise the Introduction/Discussion to more pointedly articulate how this study compares to previous ones that examine microbial community composition (including in animals), and how it raises new questions that move the field forward conceptually.

We have included more detailed metadata in our revised manuscript, and updated our analyses to include both sample volumes and DNA concentrations. We have revised our Introduction and Discussion to more directly compare our study with other studies, targeting research investigating microbial community composition in animals and focusing on other convergently evolved organisms. We have carefully articulated how our study is a conceptual advance, relevant to the fields of evolution, ecology, and microbiology.

Reviewer #1:See summary commentsReviewer #2:This paper presents a compelling combination of observational data and experiments to test the hypothesis that distantly-related families of pitcher plants share a common core microbiota independent of their geographic locations and evolutionary histories. The authors first present data supporting the hypothesis that the inquiline microbial communities of Nepenthes and Sarracenia species are more similar to one another than they are to their external environments. Such similarity across these huge geographic distances suggests that the plants' leaf environments selectively enrich a similar suite of inquiline taxa and metabolic pathways. The authors then conduct a clever field experiment wherein Nepenthes host plants, which evolved in Asia, are brought into a Sarracenia habitat in New England and are allowed to naturally develop their microbiota. The authors found that Nepenthes pitcher community composition matched that of local Sarracenia pitchers, suggesting 1) that some shared features of these two families deterministically influenced the success of particular clades of micro-organisms, and 2) that differences between pitcher families' microbiota are not evolutionarily conserved, and are instead products of their local source pools. In terms of novelty, a number of other studies have previously documented convergent interactions among hosts and their microbial symbionts (e.g., N-fixing and mycorrhizal associations in plants, diet-driven gut microbiota in animals), but this is among the first to identify shared communities that encompass such a wide taxonomic breadth (arthropods, micro-eukaryotes, bacteria). It is also one of the first to experimentally test how these communities form when placed well outside of their natural ranges (though analogous 'natural experiments' have been conducted on Sarracenia introduced to Switzerland by Fragnière 2012 and Zander et al., 2016, which might be discussed). The inclusion of metagenomic data is another valuable contribution. Few studies have done so, and none as thoroughly as this one.

We appreciate your clear and concise summary; it highlights what we hoped readers would take from the manuscript.

We have added a sentence to the Introduction describing the Swiss *Sarracenia purpurea*: “Various arthropods have co-diversified with their pitcher plant hosts, suggesting ecological dependence and a shared evolutionary history (Satler et al. 2016). In fact, even though the species *S. purpurea* was introduced to Europe over 100 years ago, it houses very few insect inquilines, as compared to plants in native habitats (Zander et al. 2016); the close associations of pitchers and arthropods may be slow to evolve.” We have not included the reference in our Discussion as the Swiss plants are growing in a place that does not have any native pitcher inquilines – an aspect of natural history which translates to a very large difference between the Swiss circumstance and our manipulative experiment, in which the *Nepenthes* were placed into a *Sarracenia* habitat and exposed to the same pool of organisms already colonizing the convergently evolved Sarracenia.

I outline a few concerns with the current draft below.One major issue with the study design is that it appears pitcher leaves within each species were sampled at a single point in time from their respective locations. Given that pitcher environments and microbial communities have been observed to change markedly in response to season and prey deposition (e.g., Koopman et al., 2010; Sirota et al., 2013 PNAS) it would be very helpful to also have data on prey biomass (mg prey/ml), oxygen concentration, and (most importantly) living microbial and arthropod biomass in each pitcher. Were these data collected? How might differences in prey biomass influence the results?

We agree that successional and seasonal dynamics can affect the communities within pitchers. But in Singapore, we sampled opportunistically at the same sites in different years (often traveling in different months of the different years), and saw almost no differences in our diversity measures among years (see our paper: Baker et al., 2016, Phil. Trans. B.). Much of our fieldwork took place in Southeast Asia. In remote locations, when it was impossible to specifically age pitchers or visit twice, sampling across pitchers was the best option: when we could only sample at one time point, we purposely sampled from pitchers at different successional stages in order to recover the greatest possible diversity. We recognize there can also be seasonal differences in diversity, but our data suggest successional stage and the randomness of prey capture have larger influences on microbiome diversity than season, and we are confident our sampling design reflects real differences between the biodiversity of different pitcher plant species across time and space.

We have included a discussion of oxygen concentrations in our Discussion (second paragraph). Oxygen is likely to be an unmeasured variable in our study; we did try to measure oxygen concentrations in the field, but our initial tests with a portable instrument proved the instrument was unreliable and we stopped collecting these data.

With regard to prey biomass and living microbial/other biomass, DNA concentration is the best and most appropriate measure for this study. Living biomass and DNA concentrations are strongly correlated (e.g. Marstorp, Guan and Gong, 2000), and in our samples it is a better measure than cell counts because some cells are lysed in the high-salt preservative we use to collect pitcher fluids in Borneo and other distant field sites. In fact our related study (Bittleston et al., 2016a) confirms that the great majority of the DNA sequences extracted from pitcher fluid samples came from living inquilines, and not from dead prey (in other words, DNA concentrations are not measuring dead biomass). DNA concentrations are also the most appropriate because some prey break down very quickly in pitcher plants, and measurements of prey biomass can result in an underestimation of resource availability (Heard, The American Midland Naturalist 1998; and personal observations).

We have added the DNA concentration of each sample to our metadata (Supplementary Dataset 1), and included DNA concentration as a variable in our analyses of *Nepenthes* and *Sarracenia* communities (see Supplementary file 1—table S4 in our revised manuscript).

Introduction, second paragraph: This paragraph draws some parallels between microbial diversity, global change, convergent interactions that are a bit of a stretch and are not covered in the design/Results/Discussion. If the goal is to justify the study, I think it's better to scrap the narratives of microbial diversity and global change, and instead introduce the question at hand – how microbial communities assemble in highly-specialized, convergently-evolved structures. This could be done by introducing the concept of convergence in dietary structures among various clades (e.g. ruminants and the hoatzin, blood-feeding arthropods, wood-eating insects) and then introducing the potential importance of microbial digestive associates in these cases. The case can then be made that pitcher plants are uniquely situated for testing this concept.

We appreciate your suggestions for improving our framing. We decided to delete the paragraph you reference, and rewrote the Introduction to 1) address microbial community assembly in convergent hosts across distinct animal/plant systems, and 2) explicitly articulate how pitcher plants are a model of microbial community dynamics.

Introduction, third paragraph: Citations here are misleading. Why not cite the studies that have actually used pitcher plants to investigate metacommunities and community assembly? (e.g.., Kneitel and Miller 2003; Buckley et al., 2003; Armitage 2017).

We had chosen to cite general references to community assembly and metacommunity theory, but realize these citations were confusing. We have edited the citations to include your suggestions.

Introduction, third paragraph: It's not clear what these 'clear parallels' are between these different communities. Here again, citations are somewhat misleading in that they do not address the aforementioned parallels between pitcher plant and phyllosphere/nectar/rhizosphere microbionts.

Given our extensive editing, we don't feel this sentence is necessary any longer, and we have deleted it.

Introduction, last paragraph: Please explain how 'convergence' is defined. As it stands, it is a nebulous enough term that it can be used to assign meaning to a huge variety of different patterns, such as:– many shared OTUs– some shared numerically-abundant OTUs– no shared OTUs but some shared genes conferring an implied function– no shared OTUs but many shared genes conferring no implied function– shared relative abundance of OTU families but no shared OTUs– shared network of OTU co-occurrences among hostsAre there any commonly-used definitions? Is there a criterion for rejecting convergence?

Convergence is defined based on similarity, and similarity is a relative concept, making convergence inherently difficult to define precisely; nonetheless we seek precision and have written a sentence at the beginning of the results to clarify the definition we use to probe for the presence or absence of convergence: “Communities from Southeast Asian and North American pitchers were defined as converging if the communities were more similar to each other than to the communities of the environments immediately surrounding the plants, even despite the vast geographic distance between them.” Much of the conceptual work behind this definition is described in our publication Bittleston et al. (2016b); in this publication we describe in detail our use of the word convergence. We have more explicitly referenced this publication in our revised manuscript. Using this definition, our study finds convergence as measured by alpha diversity (based on both observed OTUs and Shannon diversity), and as measured by the phylogenetic groups that are most common and relatively abundant within pitchers. Using this definition, tests of convergence can definitely fail. In terms of beta-diversity, our test of convergence is inconclusive, because in Figure 1Ci and Cii, the *Nepenthes* and *Sarracenia* communities do not cluster more closely with each other than they do with the communities from their surrounding environments. But the results of our manipulative field experiment suggest *Nepenthes* and *Sarracenia* pitchers function as similar selective environments, when exposed to the same microbial pool; the same kinds of communities assembled in each genus in the common habitat.

Also, if the authors are truly searching for evidence of convergent interactions, then why aren't interactions being measured? The authors are making the tacit assumption that shared communities equates to shared interactions within those communities, which is not necessarily the case.

We would argue that we *are* measuring interactions: the interactions between the pitcher plants and the microcosms assembling within them. While exploring a subset of specific two-way interactions using more traditional ecological approaches emerges as a logical next step, it was not the focus of our paper, and could not be; we need the knowledge represented by the current manuscript to move forward. Focusing on a subset of interactions might also result in a different bias (how would we know the subset is representative?), and given that our aim was to work across the East Coast of North America, in Singapore, and in Borneo, high throughput molecular methods were definitely the first priority.

Subsection “Relocated Nepenthes converge on Sarracenia-like communities”, last paragraph: Were pitcher leaves more similar to tubes with prey than they were to tubes without prey? These comparisons could identify how much variation prey capture accounts for, independent of the pitcher leaf.

Thank you for this suggestion; you make a good point. We have now included the two types of tubes as different symbols in Figure 4Di and Dii. We also added text to the Results: “NMDS plots indicate glass tubes with added prey did not assemble communities more similar to the experimental pitcher communities than glass tubes without added prey.”

Subsection “Relocated Nepenthes converge on Sarracenia-like communities”, last paragraph: Given the differences in sample sizes, it's entirely conceivable that more tube samples would recapitulate the observed variation among pitchers and the 'reduced variation' would vanish. Therefore, I suggest removing the statement that the glass communities were different from the experimental pitchers owing to their variation. Also, I couldn't find the PERMANOVA results for experimental pitchers vs. glass tubes. The negative results should also be included for the sake of transparency.

We have removed the statement, and have included the analysis of pitchers vs. glass tubes in Supplementary file 1—table S4. We also added text to the Results: “Only a very small portion of the variation in community composition was explained in analyses of pitchers vs. glass tubes; however, the differences were highly significant (bacteria: *envfit* R^2^ = 0.05, P < 0.001, *adonis* R^2^ = 0.02, P < 0.001; eukaryotes: *envfit* R^2^ = 0.09, P < 0.001, *adonis* R^2^ = 0.03, P < 0.001). The analysis suggests a sterile, pitcher-shaped form is almost, but not quite entirely, sufficient for acquiring a pitcher plant-like microcosm (Figure 4D).”

Discussion, second and fifth paragraphs: Without data on the actual contribution of microbes to host functioning (e.g., via prey decomposition/isotope uptake paired with fluid dilution/antibiotic treatment), calling these 'convergent interactions' remains speculative, as the term 'interaction' implies some type of reciprocal responses between the host organism and its microbiota. However, we already know that removing of many components of the pitcher food web (such as micro-arthropods) carries no perceptible consequences for their hosts (Butler, Gotelli and Ellison, 2008; Baiser, Ardeshiri and Ellison, 2011). Again, a more formal introduction to 'convergent interactions' would be welcome.

Please also see our comments, above. We agree that direct evidence of reciprocal responses between the host organism and its microbiota would be strong evidence of convergent interactions. However, without our current data, the design of these manipulative experiments would be difficult to justify, and difficult to make happen in remote habitats: hopefully these kinds of experiments will take place in the near future. Meanwhile, our large dataset, collected from two continents, also builds a compelling case for convergent interactions: we have clearly demonstrated that two distantly related, convergently evolved genera of pitcher plants select for similar communities, with microbiomes with increased degradation capabilities compared to other habitats. However, to more appropriately reflect what we have learned, we have added “appear” to modify our updated statement: “Convergently evolved pitchers appear to cause convergent interactions (Bittleston et al., 2016b) between the two genera and their associated pitcher microcosms.”

Subsection “Experimental relocation of Nepenthes spp. to a New England bog”, second paragraph: I'm not convinced that the control groups (bog water, soil) are appropriate references to use for detecting convergence among pitchers. For instance, bog and soil water are treated as "null" environments to which pitcher leaves are compared. However, since pitcher leaves are really being colonised by microbes derived primarily from the guts and bodies of trapped or resident arthropods, then wouldn't those organisms' combined microbiota serve as a more meaningful source pool? That way, the resulting pitcher communities could be directly related to environmental filtering from the pool of colonists. What if the insect prey microbiota ended up being identical to the pitcher microbiota?

Although insect gut microbes may be one source of pitcher microbiota, we do not agree they are the primary colonizers: the main bacterial groups we find in pitchers are ones that are generally soil-, water-, and plant surface-associated. Sirigusa et al. (2007) suggest that bacteria cultured from *Sarracenia* pitchers are of insect origins, however, the taxa referenced (e.g. *Serratia, Pantoea*, etc.) are also common soil and freshwater habitats. Takeuchi et al. (2015) found pitcher plant bacteria were more similar to phyllosphere bacteria than gut bacteria. In one of our own unpublished studies, we excluded insects (both prey and inquilines) from *Sarracenia purpurea* pitchers and still found common pitcher microbes colonizing the pitchers, even when insects were absent. Of course, ideally, we would have the entire spectrum of potential microbiota from all potential source environments as our “null” environments, but this kind of sampling simply wasn't feasible; based on what we knew, bog water and soil emerged as more relevant than insect guts. Because almost all of the microbes we found in our pitcher samples were also present (albeit at very low relative abundances, sometimes) in our environmental samples (see Figure 2), we feel that the sampled environments do represent good “null” comparisons for this study.

Subsection “Analyses of 16S and 18S diversity”, third paragraph: Unweighted UniFrac is notoriously sensitive to rare members of the community. Please also include similar weighted UniFrac or Bray-Curtis based dissimilarity analysis in the supplement so that readers can assess the influence of rare vs. dominant taxa on beta-diversity.

Thank you for your suggestion. We have run our beta-diversity analyses with Bray-Curtis as well and included the Results in our Supplementary file 1—table S4.

Subsection “Analyses of 16S and 18S diversity”, last paragraph: How were these trees generated? I can't assess their quality without more information such as the identification of major clades on the trees, some branch length scale, bootstrap values, rooting methodology, etc.

We did not include branch lengths in Figure 2 because visually, we find it easier to differentiate between clades when branch lengths are omitted. But to facilitate a more straightforward evaluation of the trees, we have added bootstrap value indicators in Figure 2, and have also included new trees as a figure supplement to Figure 2 with both branch lengths and branch length scales. We have also added text detailing rooting methodology to the Materials and methods section: “The bacterial tree was rooted with Archaea, and the eukaryotic tree was rooted in Streptophyta (land plants and most green algae).”

Reviewer #3:[…] 1) My most substantive comment is that I cannot find evidence in the Materials and methods or supplementary material that the authors standardized sample volume between the pitchers and the other environmental samples (soil and bog water). Since the authors use species accumulation curves (Figure 1B) to suggest that pitchers are more species depauperate than surrounding soil or bog water, standardizing sample volume is critical. Specifically, if there is any micro-scale spatial turnover in species composition, collecting a large sample, homogenizing it, and extracting 4000 sequences will result in capture of more OTUs that collecting a smaller sample, homogenizing it and extracting 4000 sequences.

Thank you for pointing this out: we have added sample volume metadata for our environmental samples to Supplementary file 1—table S3, and show that average sample volumes are similar across different types of samples (Supplementary file 1—table S3). We also tested for a relationship between sample volume and alpha diversity, using two different methods: we checked for a correlation between observed OTUs and volume (and found no correlation), and we re-ran our analysis of Shannon diversity using a subset of 155 samples that were all extracted around the same time, in exactly the same way, and with the same extraction volume. We see the same patterns of Shannon diversity in this subset as we see in our larger dataset. We are confident that sample volume is not driving the patterns of Figure 1Bi and Bii.

2) In the analysis of within-genera correlates of community structure (subsection “Within each genus, pitcher species, acidity, form, and volume correlate with community composition”, third paragraph), it is not clear if you have controlled for species. If not, any community correlations with the physical and chemical properties of pitchers would be psuedoreplicated as these physical/chemical differences could simply stand in for species differences.

Thank you for raising this issue. For the within-genus correlates that we examine, it would be difficult to control for host species within the multivariate analyses. For pH, the best way to control for host species is to examine the effect of pH within each of the three *Nepenthes* species that have broad pH ranges (*N. gracilis, N. rafflesiana* and *N. stenophylla*). We have added these analyses for these three host species, and for each we find that pH still has a very strong correlation with community diversity (Supplementary file 1—table S4). These new analyses clearly show that the strong correlation with pH is not simply a result of the different pH of different host species. Unfortunately, for the plant form analysis in *Sarracenia*, the different forms do not occur within a single species, and we cannot separate these factors. We have added a sentence to the manuscript explaining that plant form is confounded with host species and have removed plant form from our analyses.

3) In the gene pathways analysis, I couldn't determine if you were using samples from the same region (transplant experiment) or different regions (observation dataset). This matters in the interpretation of the results.

We are using samples from different regions (the observation dataset), and definitely want this point to be clear to all readers. We have added text that says the metagenomes are from “field collected (not experimentally relocated)”pitchers.

4) There has been a fair amount of work comparing the strength of habitat filtering on macroscopic organisms vs. microbial, for example in lakes and bromeliads. Your study seems well positioned to follow up these ideas, in particular Figure 2 suggests that the pitcher habitat filters bacteria and eukaryotes to different degrees. Can you incorporate in your Discussion?

Thank you for the interesting connection you make for us. We have added a sentence to our Discussion: “Fewer eukaryotes consistently colonize *Nepenthes* or *Sarracenia* pitchers, as compared to bacteria (Figure 2); the pattern is consistent with a stronger habitat filtering with increasing body size, as observed in bromeliad phytotelmata (Farjalla et al., 2012).”